# Analysis and Correction of Polarization Response Calibration Error of Limb Atmosphere Ultraviolet Hyperspectral Detector

**DOI:** 10.3390/s22218542

**Published:** 2022-11-06

**Authors:** Haochen Li, Zhanfeng Li, Yu Huang, Guanyu Lin, Jiexiong Zeng, Hanshuang Li, Shurong Wang, Wenyao Han

**Affiliations:** 1Changchun Institute of Optics, Fine Mechanics and Physics, Chinese Academy of Sciences, Changchun 130033, China; 2University of Chinese Academy of Sciences, Beijing 100049, China; 3Guangdong-Hong Kong-Macao Joint Laboratory for Intelligent Micro-Nano Optoelectronic Technology, Foshan University, Foshan 528000, China

**Keywords:** polarization response, Muller matrix, Stokes vector, amplitude attenuation coefficients

## Abstract

A UV hyperspectral instrument was designed with a polarization measurement channel for real-time in-orbit polarization correction to reduce the influence of polarization on the detection accuracy of atmospheric radiation. One of the prerequisites for in-orbit polarization calibration is accurately calibrating the instrument’s polarization properties in the laboratory. This study first introduces the calibration method and measuring device of the polarization characteristics of the ultraviolet (UV) hyperspectral detector and conducts a polarization calibration test of the instrument. The two main error sources introduced by the calibration device were emphatically analyzed, and the correction method of the error sources was deduced theoretically. Finally, the polarization calibration results of the UV hyperspectral detector were corrected, and the uncertainty analysis of the corrected calibration results was about 1.4%, which provides effective ground polarization calibration data for the on-orbit polarization correction of the instrument.

## 1. Introduction

The Ultraviolet Hyperspectral Ozone Detector is an atmospheric trace gas detection instrument that will be carried on the Fengyun 3(FY-3) meteorological satellite. It will obtain the solar backscattering spectral radiation data in the ultraviolet-visible wavelength band of the atmosphere through the limb observation method. The schematic diagram of the edge detection is shown in Figure 1. And retrieve the vertical distribution information of trace gases such as global ozone ($O_{3}$), nitrogen dioxide ($NO_{2}$ ) and sulfur dioxide ($SO_{2}$). The gas information will be further applied to related research on climate change, atmospheric chemistry and atmospheric environment. The datasets acquired by the Ultraviolet Hyperspectral Ozone Detector will help to quantify the complex interactions between natural and human activities, climate, atmospheric composition, and associated chemical and physical processes such as the impact of human activities and natural processes on tropospheric ozone (the precipitous loss of Antarctic and Arctic stratospheric ozone) and air quality.

Due to the presence of polarization-sensitive elements, such as scanning mirrors, diffraction gratings, and a beam splitter, the response of the Ultraviolet Hyperspectral Detector to incident light varies with the polarization state, even when the light intensity remains constant [1,2,3,4]. As a result, the detection accuracy of atmospheric radiation will be affected. At present, most atmospheric remote sensing instruments use the depolarizer method to reduce the polarization sensitivity of the instrument, as shown in Table 1 [5,6,7,8,9,10,11,12].

However, for the UV Hyperspectral Ozone Detector, there are two drawbacks if depolarizers are used, which affect the instrument’s size and degrade the image quality [13]. The UV Hyperspectral Ozone Detector consists of a front scanning mirror and a rear optical system. The scanning mirror aperture is 125 × 130 mm. If a depolarizer is added in front of the scanning mirror, it is at least 130 mm. Due to the limited resources of satellites, this not only makes the instrument larger but also makes it impossible to guarantee the polarizer’s homogeneity and spatial stability. However, if the depolarizer is placed behind the scanning mirror, the depolarizer only depolarizes the rear optical system, while the front-end polarization phenomenon still affects the detection. Moreover, the addition of depolarizers introduces double image aberration, which degrades the imaging quality and impacts the spectral resolution of the entire system [14].

Therefore, the UV Hyperspectral Ozone Detector adopts an on-orbit polarization correction method to ensure the depolarization effect. The on-orbit polarization correction method depends on establishing the correlation between the polarization state of the incident light and the polarization-sensitive device. It is realized by calibrating the Mueller matrix of the instrument and by measuring the Stokes vector of polarized light [11,12,15,16,17,18,19]. The key to on-orbit polarization calibration is to determine the polarization calibration factor by ground calibration in the laboratory and on-orbit measurements by the instrument. In accordance with the requirements of the on-orbit polarization calibration method, the polarization response properties of each instrument channel, i.e., the normalized Mueller matrix elements, need to be calibrated in the laboratory. The polarization spectrum reactivity of space remote sensing instruments is a key parameter for calibrating instruments in laboratories, and the results have practical significance for assessing the instrument’s performance [20,21,22,23]. This paper introduces the laboratory calibration of the UV hyperspectral instrument’s polarization properties and implements error correction for the system’s radiation transmission link, providing high-precision ground polarization calibration information for on-orbit polarization calibration.

## 2. On-Orbit Polarization Correction Method for Ultraviolet Hyperspectral Detector

Light from the sun is unpolarized before it reaches the atmosphere, but becomes partially polarized after being scattered by atmospheric constituent gases and solid or liquid particles. The polarization state of scattered light in the atmosphere depends on the distribution of atmospheric components and solid–liquid particles [24].

In order to reduce the influence of polarization on the measurement results of the space remote sensing spectrometer, one method is to reduce the linear polarization responsivity of the instrument, that is, the optical depolarizer scheme. Another method is to measure the polarization characteristics of the instrument and the incident light wave for correction, that is, on-orbit polarization correction. Due to the limitations of the depolarizer scheme in hyperspectral detectors, it is difficult to meet the requirements of ultraviolet hyperspectral detectors, affecting the measurement accuracy of space remote sensing spectrometers. However, the on-orbit polarization correction theory can effectively reduce the influence of atmospheric polarization characteristics on instruments. The influence of the measurement accuracy and the theoretical derivation of the on-orbit polarization correction is carried out below.

The mathematical methods to describe the polarization state of light waves mainly include the trigonometric function method, the Jones vector method and the Stokes vector method [25,26]. Both the trigonometric function method and the Jones vector method describe the polarization characteristics of light waves on the basis of amplitude and phase, and can only describe the polarization characteristics of fully polarized light. The Stokes vector method is based on light intensity and can describe the polarization characteristics of any light wave polarization state. Therefore, in this paper, the Stokes vector is adopted to represent beam information, and the Muller matrix is used to describe the polarization properties of the medium. The expression is as follows:(1)$$\left[\begin{array}{c} I\\ Q\\ U\\ V \end{array}\right]=\left[\begin{array}{cccc} M_{11} & M_{12} & M_{13} & M_{14}\\ M_{21} & M_{22} & M_{23} & M_{24}\\ M_{31} & M_{32} & M_{33} & M_{34}\\ M_{41} & M_{42} & M_{43} & M_{44} \end{array}\right]\cdot \left[\begin{array}{c} I_{0}\\ Q_{0}\\ U_{0}\\ V_{0} \end{array}\right]$$

The *I* component represents the light intensity information, that is, the radiant energy passing through a unit area; the *Q* and *U* components represent the direction and intensity of linear polarization; the *V* component represents the circular polarization component [26]. Since the target of the UV hyperspectral instrument is atmospheric backward scattered light, its circularly polarized light component is neglected [27]. The fourth term of the Stokes vector is zero, and it becomes:(2)$$S=M_{11}I_{0}+M_{12}Q_{0}+M_{13}U_{0}.$$

First, it is proposed by Formula (Equation 2) that $M_{11}I_{i}$ becomes:(3)$$S_{i}=M_{11}I_{i}\left(1+\frac{M_{12}}{M_{11}}\frac{Q_{i}}{I_{i}}+\frac{M_{13}}{M_{11}}\frac{U_{i}}{I_{i}}\right)=M_{11}I_{i}\left(1+m_{2}q_{i}+m_{3}u_{i}\right)=M_{11}I_{i}\frac{1}{c_{{}_{pol}}^{i}}$$
(4)$$c_{pol}^{i}=\frac{1}{1+m_{2}q_{i}+m_{3}u_{i}}$$
(5)$$m_{2}=\frac{M_{12}}{M_{11}};m_{3}=\frac{M_{13}}{M_{11}}$$
(6)qi=QiQiIiIi,ui=UiUiIiIi,
where $c_{pol}^{i}$ is the on-orbit polarization correction factor, which is determined by the polarization characteristics of the incident light and the polarization characteristics of the space remote sensing spectrometer; $m_{2}$ and $m_{3}$ are the normalized Mueller matrix parameters of the space remote sensing spectrometer, which characterize the polarization characteristics of the instrument; $\rho_{i}$ and $\mu_{i}$ is the normalized Stokes vector parameter of the incident light, which characterizes the polarization characteristics of the incident light.

The deformation of Formula (Equation 3) can be obtained:(7)$$S_{i}c_{{}_{pol}}^{i}={S_{i}}^{\prime }=M_{11}I_{i}.$$

It can be seen from Formula (Equation 7) that only the on-orbit polarization correction factor is required—the on-orbit polarization correction can be performed on the output signal of the instrument and the corrected effective signal ${S_{i}}^{\prime }$ can be obtained, which is only related to the light intensity of the incident light wave. According to the responsivity $M_{11}$ of the space remote sensing spectrometer to unpolarized light, the light intensity $I_{i}$ of the incident light wave can be obtained.

Therefore, the core work of the on-orbit polarization correction of space remote sensing spectrometers is to solve the on-orbit polarization correction factor. From the definition Formula (Equation 4) of the on-orbit polarization correction factor, it can be known that the following two aspects are required to solve the on-orbit polarization correction factor. One is to solve the normalized Mueller matrix elements $m_{2}$ and $m_{3}$ of the space remote sensing spectrometer, which are determined by the polarization response characteristics of the instrument itself, and can be obtained by laboratory calibration, that is, the ground calibration of the polarization characteristics of the instrument. The second is to solve the normalized Stokes vector parameters $\rho_{i}$ and $\mu_{i}$ of the incident light wave, which are determined by the polarization state of the incident light wave and are obtained by on-orbit measurement, that is, the on-orbit measurement of the polarization characteristics of the incident light wave. This paper mainly focuses on the first aspect of the work, and the following will focus on the ground calibration of space remote sensing spectrometers.

## 3. Ground Calibration of the Polarization Spectrum Responsivity of the UV Hyperspectral Instrument

### 3.1. Instrument Description

The UV Hyperspectral Detector is a limb viewing UV/visible spectrometer. The optical system of the spectrometer has three channels. The input beam undergoes two splits—first it is divided between the main science channel and the polarization measurement channel (PMC, channel 3 in Figure 2). The main science channel is then divided into two working spectral channels (channel 1 and channel 2 in Figure 2). The instrument works in the spectral range 290–500 nm, with a spectral resolution of 0.6 nm and a spatial resolution of 100 km × 3 km.The main detection target of the instrument is the spectral radiance of the 15–60 km limb atmosphere. The schematic diagram of limb scanning detection is shown in Figure 3. The main parameters of the instrument are shown in Table 2.

In order to meet the high-precision detection of atmospheric signals in the ultraviolet band of 290–500 nm, the Ultraviolet Hyperspectral Detector adopts a double monochromator structure, which is mainly composed of a telescope and a dispersive imaging system. Its optical system is shown in Figure 2. In order to improve the utilization rate of light energy, the off-axis parabolic mirror is used in the telephoto system. The reflection system has no chromatic aberration, and the off-axis paraboloid can correct spherical aberration [28]. Atmospheric spectral radiation in the Earth’s limb direction is focused on the pre-dispersion system by a scanning mirror and telescope system. The pre-dispersion prism expands the spectrum of the incident beam in space, and is divided into the main scientific channel and the polarization channel through the incident slit. The main scientific channel is then divided into two spectral channels by the beam splitter. In the polarization characteristic ground calibration, the output signals of channel 1 and channel 2 are used to calibrate the polarization response of the instrument. The secondary dispersion system with working wavelengths of 290–400 nm and 390–500 nm includes a collimating mirror, a plane reflection grating (1800 L/mm) and a Petzval imaging lens group. The formed spectral image is collected by the Petzval objective lens on the photosensitive surface of the detector. The limb detection signal enters the telescopic system and the spectrometer system through the scanning mirror (the scanning mirror is not shown in Figure 2). Along the column direction of the area array detector, it represents the change of the atmospheric radiance of the Earth’s limb in the specified area with wavelength, that is, the spectral dimension. The detector uses CMOS and is cooled to −10 °C with a semiconductor cooler.

The main science channel and the polarization measurement channel (PMC) share the telescopic system, the entrance slit and the pre-dispersing prism. Passing through a pre-dispersing prism while simultaneously causing a portion of the beam to be incident on the PMC at Brewster’s angle produces an almost fully polarized beam with the polarization directed towards the paper (represented by the intersecting circles), that is, parallel to the instrument slit. After an internal reflection, it leaves the prism and is refracted into the polarization system. The optical structure of the PMC is shown in Figure 4.

It is important to carry out the ground polarization calibration and methodologies investigation for the two working spectral channels to achieve in-orbit high-precision polarization calibration.

### 3.2. Calibration Method

According to Formula (Equation 2), when the incident light’s wavelength and the angle of incidence are fixed, the elements of the Mueller matrix of the instrument $M_{11}I_{0}$, $m_{2}$ and $m_{3}$ are constant. Therefore, to solve the three unknown quantities, at least the output signal of the instrument needs to be measured under the incident conditions of light waves of three different polarization states.

In order to eliminate the influence of a single measurement error on the measurement results, multiple measurements were performed, and the solution was obtained using overdetermined equations. Specifically, the linearly polarized light was generated by a wire grid polarizer; the linearly polarized light with different polarization states was generated by changing the polarization azimuth $\eta_{i}$ of the polarizer.
(8)$$\begin{array}{c} S_{i}=M_{11}I_{0}+M_{12}Q_{i}+M_{13}U_{i}\\ =1\cdot (M_{11}I_{0}+\cos2\eta_{i}\cdot \left(M_{12}I_{0}\right)+\sin2\eta_{i}\cdot \left(M_{13}I_{0}\right). \end{array}$$

The device acquires a signal for every 15° rotation of the polarization azimuth angle. From 0° clockwise rotation to 360°, the device acquired a total of 25 signals; thus, combining the measuring results for those 25 signals, the following equations are obtained:(9)$$\left[\begin{array}{ccc} 1 & \cos2\eta_{1} & \sin2\eta_{1}\\ 1 & \cos2\eta_{2} & \sin2\eta_{2}\\ \vdots & \vdots & \vdots \\ 1 & \cos2\eta_{25} & \sin2\eta 25 \end{array}\right]\left[\begin{array}{c} M_{11}I_{0}\\ M_{12}I_{0}\\ M_{13}I_{0} \end{array}\right]=\left[\begin{array}{c} S_{1}\\ S_{2}\\ \vdots \\ S_{25}, \end{array}\right]$$
where i=1,2,…,25; $\eta_{i}=15\cdot {\left(i-1\right)}^{{}^{\circ}}$; then, the normalized Mueller matrix elements $M_{11}I_{0}$, $m_{2}$ and $m_{3}$ of the UV hyperspectral instrument can be calculated. The polarization calibration test setup of the UV hyperspectral instrument was established in accordance with the above theoretical analysis.

### 3.3. Experimental Setup

The polarization properties calibration device consists of a xenon lamp, UV collimator, wire grid polarizer, and a UV hyperspectral instrument that emits “white light” at a high color temperature of 6000 K, which is close to that of sunlight and covers a broad continuous spectrum (185 nm to 2000 nm) from the UV to infrared regions.The stability of the output light is about 0.2%, and the typical drift value is ±0.5%/h. The graph of the spectral irradiance of xenon lamp as a function of wavelength is shown in Figure 5. The UV collimator was independently developed. In order to ensure the ultraviolet efficiency, a plane mirror and an off-axis parabolic mirror structure were used, and both mirrors were coated with Al+MgF2 film.

Since the calibration of the polarization characteristics of the instrument requires the incident light to be linearly polarized light, and the effective aperture is greater than 40 mm, the crystal polarizer cannot meet the coverage of the full aperture, so the wire grid polarizer was used as the polarized light output. When light is incident on the wire grid, the P light will encounter the dielectric and is transmitted, while the S light will encounter the reflector and is reflected. Changes in the light’s azimuth angle determined the angles at which the wire grid polarizer was set on a two-dimensional rotation table [29].

The wire grid polarizer was an Edmund ultra-wideband wire grid polarizer, its linear degree of polarization in the range of 290–500 nm is greater than 99%, The wire grid polarization characteristic curve is shown in Figure 6. and the incident angle is less than ±20°. The rotation table was the SOFN INSTRUMENTS CO. motorized rotation table with a precision of rotation angle higher than 0.01°. The experimental setup is shown in Figure 7.

The calibration test diagram of polarization characteristics of the Ultraviolet Hyperspectral Detector is shown in Figure 8.

The polarization response calibration results of the UV Hyperspectral Detector are shown in Figure 9. The scanning mirror angle was 13.6°.

## 4. Error Source Analysis of Laboratory Calibration of the Polarization Properties

It is clear from an analysis of the Mueller matrix calibration method’s radiation transmission link that the polarization response of the UV collimator and the wire grid polarizer to incident light are the sources of the method’s main errors [30].

### 4.1. Effect of UV Collimator on the Uniformity of Output Light Intensity

The UV collimator has a reflecting mirror as part of its internal design that is polarization-sensitive. Then, the unpolarized light generated by the xenon lamp turns into partially polarized light after passing through the collimator. The output linear polarized light intensity is a constant when the unpolarized light passes through the linear polarizer at a different polarization angle. In this case, the Stokes vector of the output light passing through the wire grid polarizer is the matrix multiplication of the Mueller matrix of the wire grid polarizer and the Stokes vector of the incident light. So, the Mueller matrix *M* of a perfect wire grid polarizer is [27,31,32]:(10)$$M_{p}=\frac{1}{2}\left[\begin{array}{ccc} 1 & \cos2\beta & \sin2\beta \\ \cos2\beta & \cos^{2}2\beta & \cos2\beta \sin2\beta \\ \sin2\beta & \cos2\beta \sin2\beta & \sin^{2}2\beta , \end{array}\right]$$
where β is the polarization azimuth angle. Suppose the degree of polarization of the outgoing light passing through the collimator is *P*, then the partially polarized light can be decomposed into:(11)$$\begin{array}{c} \left[\begin{array}{c} I_{o}\\ Q_{o}\\ U_{o} \end{array}\right]=\left[\begin{array}{c} \left(1-P\right)I_{o}\\ 0\\ 0 \end{array}\right]+PI_{o}\left[\begin{array}{c} 1\\ \cos2\beta_{o}\\ \sin2\beta_{o} \end{array}\right]=I_{o}\left[\begin{array}{c} 1\\ P\cos2\beta_{o}\\ P\sin2\beta_{o}. \end{array}\right] \end{array}$$

The Stokes vector of the output light from a perfect wire grid polarizer:(12)$$\begin{array}{c} \left[\begin{array}{c} I^{\prime }\\ Q^{\prime }\\ U^{\prime } \end{array}\right]=M_{p}I_{0}\left[\begin{array}{c} 1\\ P\cos2\beta_{o}\\ P\sin2\beta_{o} \end{array}\right]=\frac{I_{0}}{2}\left(1+P\cos2\left(\beta -\beta_{o}\right)\right)\left[\begin{array}{c} 1\\ \cos2\beta \\ \sin2\beta , \end{array}\right] \end{array}$$
where $\beta_{o}$ is the angle between the polarization direction of the incident light and the 0° polarization angle of the polarizer. When the polarization azimuth angle β = $\beta_{o}$ (the polarization direction of the rotating linear polarizer is the same as the polarization direction of the incident light), the output light intensity is the largest, which is:(13)$$I_{\max}^{\prime }=\frac{I_{0}}{2}\left(1+K\right).$$

When $\eta =\eta_{0}+\pi /2$ (the polarization direction of the rotating linear polarizer is perpendicular to the polarization direction of the incident light), the output light intensity of linear polarization is the minimum, which is:(14)$$I_{\min}^{\prime }=\frac{I_{0}}{2}\left(1-K\right).$$

Taking the average value as the actual value, the maximum and minimum errors are:(15)$$\bar{\left(I^{\prime }\right)}=(I_{\max}^{\prime }+I_{\min}^{\prime })/2=\frac{I_{0}}{2}$$
(16)Δmax=I′maxI′max(I′)¯(I′)¯−1=P
(17)Δmin=I′minI′min(I′)¯(I′)¯−1=−P.

When the output light of the UV collimator is a partially polarized light with a degree of polarization *P*, the measurement of the instrument’s polarization properties has an error of ±P.

### 4.2. Effect of Wire Grid Polarizer on the Polarization Response of Light

Any polarizer can be considered an attenuator. When light is incident on the polarizer, the two perpendicular electric field components are attenuated differently. Under ideal conditions, the wire grid polarizer completely attenuates the light in one direction and allows the orthogonal light to pass through it, thus resulting in linearly polarized light. However, in reality, a linear polarizer cannot completely attenuate the light in a given direction. In that way, the extinction ratio and transmittance are used to evaluate the performance of a polarizer [33].

Suppose the two components of the electric field $E_{\|}$ and $E_{⊥}$ of the light incident to the wire grid polarizer turn into $E_{\|}^{\prime }$ and $E_{⊥}^{\prime }$ after passing through the wire grid polarizer, where ${}_{\|}$ represents the orthogonal direction and ${}_{⊥}$ represents the horizontal direction [34].
(18)$$\begin{array}{c} {E^{\prime }}_{\|}=p_{\|}E_{\|};\;0\leq p_{\|}\leq 1 \end{array}\;{E^{\prime }}_{⊥}=p_{⊥}E_{⊥};\;0\leq p_{⊥}\leq 1,$$
where $p_{\|}$ and $p_{⊥}$ represent the amplitude attenuation coefficients of the wire grid polarizer in the horizontal and vertical directions, respectively. The generalized Muller matrix of wire grid polarizers is:(19)$$M\left(\beta \right)=\frac{1}{2}\left[\begin{array}{ccc} p_{\|}^{2}+p_{⊥}^{2} & (p_{\|}^{2}-p_{⊥}^{2})\cos2\beta & (p_{\|}^{2}-p_{⊥}^{2})\sin2\beta \\ (p_{\|}^{2}-p_{⊥}^{2})cos2\beta & \cos^{2}2\beta +2p_{\|}p_{⊥}\sin^{2}2\beta & {\left(p_{\|}-p_{⊥}\right)}^{2}\sin2\beta cos2\beta \\ (p_{\|}^{2}-p_{⊥}^{2})\sin2\beta & {\left(p_{\|}-p_{⊥}\right)}^{2}\sin2\beta cos2\beta & \sin^{2}2\beta +2p_{\|}p_{⊥}\cos^{2}2\beta . \end{array}\right]$$

The Stokes vector of the output light passing through the wire grid polarizer is:(20)$$\left[\begin{array}{c} I^{\prime \prime }\\ Q^{\prime \prime }\\ U^{\prime \prime } \end{array}\right]=\frac{1}{2}M\left(\beta \right)\cdot I_{0}\left[\begin{array}{c} 1\\ P\cos2\beta_{o}\\ P\sin2\beta_{o} \end{array}\right]$$
(21)$$I^{\prime \prime }=\frac{I_{0}}{2}\left(\left(p_{\|}^{2}+p_{⊥}^{2}\right)+P\left(p_{\|}^{2}-p_{⊥}^{2}\right)\cos2\left(\beta -\beta_{0}\right)\right).$$

The relative error in the outgoing light intensity for light passing through a non-ideal wire grid polarizer was determined using the above measurement results as the actual values and is given by:(22)$$\begin{array}{c} \delta =\frac{\left|I^{\prime }-I^{\prime \prime }\right|}{I^{\prime }}=\frac{\left|\frac{I_{0}}{2}\left(\left(p_{\|}^{2}+p_{⊥}^{2}\right)+P\left(p_{\|}^{2}+p_{⊥}^{2}\right)\cos2\left(\beta -\beta_{0}\right)\right)-\frac{I_{0}}{2}\left(1+P\cos2\left(\beta -\beta_{o}\right)\right)\right|}{\frac{I_{0}}{2}\left(1+P\cos2\left(\beta -\beta_{o}\right)\right)}\\ \;=1-(p_{\|}^{2}+p_{⊥}^{2}). \end{array}$$

Therefore, when the attenuation coefficient of the wire grid polarizer is $p_{\|}^{2}+p_{⊥}^{2}\neq 1$, the relative error introduced to the measurement results on the normalized Mueller matrix of the instrument is $1-(p_{\|}^{2}+p_{⊥}^{2})$.

## 5. Detection and Correction of Polarization Measurement System

Based on the error source analysis, detection and correction of laboratory polarization calibrations were conducted for the two influencing factors introduced by the UV collimator and the wire grid polarizer to improve the calibration accuracy of polarization properties.

### 5.1. Muller Matrix of the UV Collimator Correction

The optical elements in the collimator convert the unpolarized light from the xenon lamp into partially polarized light. The collimator consists of a reflector and an off-axis parabolic mirror, and the diaphragm of the field-of-view limits the angle of the outgoing light to 50°. According to Fresnel’s formula, the reflector changes the original polarization state of the light. The effect of the off-axis parabolic mirror on the changing polarization state of the outgoing light is insignificant and can be neglected. Therefore, the Mueller matrix of the reflector in the collimator can be calculated. The optical structure diagram of the collimator is shown in Figure 10.

The Al+MgF2 complex film layer is coated on the reflector inside the collimator [35,36]. The reflectivities $R_{s}$, $R_{p}$ of s-polarized light and p-polarized light are as follows:(23)$$R_{s}={R_{s}}_{{}_{Air-MgF_{2}}}+{T_{s}}_{{}_{Air-MgF_{2}}}{R_{s}}_{{}_{MgF_{2}-Al}}{T_{s}}_{{}_{Air-MgF_{2}}}$$
(24)$$R_{p}={R_{p}}_{{}_{Air-MgF_{2}}}+{T_{p}}_{{}_{Air-MgF_{2}}}{R_{p}}_{{}_{MgF_{2}-Al}}{T_{p}}_{{}_{Air-MgF_{2}}},$$
where $R_{s}$ is the reflectivity of the s-wave, $R_{p}$ is the reflectivity of the p-wave, $T_{s}$ is the transmittance of the s-wave, and $T_{p}$ is the transmittance of the p-wave. Air-MgF2 means the light wave passes through the air in MgF2; MgF2-Al means the light wave passes through the Al in MgF2. The complex refractive index is represented by $\tilde{n}=n+ik$, where *n* represents the refractive index and *k* represents the absorptivity. The absorptivity and refractive index of Al and MgF2 in the wavelength range of 290–500 nm are shown in Figure 11.

Neglecting the effect of off-axis parabolic mirrors on the polarization of the system, the Muller matrix of the UV collimator is obtained based on the Fresnel formula [37]:(25)$$M_{c}=\frac{1}{2}\left[\begin{array}{c} R_{s}+R_{p\;}\;\;R_{s}-R_{p\;}\;\;\;\;\;\;\;\;0\;\;\;\\ R_{s}-R_{p\;}\;\;R_{s}+R_{p\;}\;\;\;\;\;\;\;\;0\\ \;\;\;\;0\;\;\;\;\;\;\;\;\;\;\;0\;\;\;\;2\sqrt{R_{s}R_{p\;}}.\cos\delta \end{array}\right]$$

When the incident angle is 50°, the curves of reflectivity, Rs and Rp, with respect to wavelength in the range of 290–500 nm, are calculated for the complex film according to Fresnel’s formula in Figure 12 [34,38].

Thus, the Muller matrix of three channels is derived. When the incident light is unpolarized, the Stokes vector of the outgoing light passing through the UV collimator is:(26)$$\left[\begin{array}{c} I^{\prime }\\ Q^{\prime }\\ U^{\prime } \end{array}\right]=M_{c}\left[\begin{array}{c} I_{0}\\ 0\\ 0 \end{array}\right]=\frac{I_{0}}{2}\left[\begin{array}{c} R_{s}+R_{p\;}\\ R_{s}-R_{p\;}\\ 0. \end{array}\right]$$

The calculated polarization degree of the UV collimator is shown in Figure 13.

The process of the xenon lamp light source passing through the collimator is represented by multiplying the Mueller matrix of the collimator with the Stokes vector of unpolarized light, that is, the corrected Stokes vector of the output light.

### 5.2. Detection and Correction of Polarization Attenuation of the Wire Grid Polarizer

The ratio of the brightness of the outgoing light to the incident light is the transmission *T* of the wire grid polarizer:(27)$$T=\frac{I(\beta )}{I_{0}}={p_{\|}}^{2}\cos^{2}\beta +{p_{⊥}}^{2}sin^{2}\beta .$$

When $\beta =0^{{}^{\circ}}$ and $\beta =90^{{}^{\circ}}$, the maximum and minimum transmittance $T_{\max}$ and $T_{\min}$:(28)$$T\left(0^{{}^{\circ}}\right)={p_{\|}}^{2}=T_{\max}$$
(29)$$T\left(90^{{}^{\circ}}\right)={p_{⊥}}^{2}=T_{\min}.$$

The maximum transmittance is achieved when the polarizer is aligned with the polarization axis, and the minimum transmittance is obtained by rotating the polarizer by 90°; their ratio is extinction ratio τ is:(30)$$\tau =\frac{T_{\max}}{T_{\min}};\;\infty >\tau \geq 1.$$

The xenon lamp light source will emit parallel light after passing through the collimator. The Rochon polarization splitting prism in the UV2000 vacuum UV spectrophotometer acts as a polarizer with a high UV transmittance, and its degree of polarization is more than 99.9%. The analyzer is a wire grid polarizer, the detector is a silicon photodetector, and a CAS spectrometer collects the output signal of the detector [39]. The setup diagram is shown in Figure 14.

The coordinate system was chosen as shown in the figure; perpendicular to the laboratory table, facing upward is the positive direction of the y-axis, and the direction in which the light propagates along the optical axis is the positive direction of the x-axis. The positive direction of the z-axis (facing towards the direction of light emission, the horizontal direction to the left is the positive direction of the z-axis, that is, the polarization azimuth angle is 0°) is determined by the right-hand rule [29,40].

First, the Rochon polarization splitting prism was placed parallel to the z-axis; light was emitted as linearly polarized light at the azimuth angle of 0°. The output signal $S_{0}$ was measured without the wire grid polarizer. Then the wire grid polarizer was inserted for the polarization axis to be parallel to the z-axis direction.

Using the azimuth angle of the wire grid polarizer to rotate 30° to measure once, from 0° to 720°, the instrument collected a total of 25 signals. The output signal is $S_{i}^{\prime }$. The measurement results are shown in Figure 15. Thus, the transmission ratio $T(\beta_{i})$ is:(31)$$T\left(\beta_{i}\right)=\frac{S_{i}^{\prime }}{S_{0}}={p_{\|}}^{2}\cos^{2}\beta_{i}+{p_{⊥}}^{2}sin^{2}\beta_{i}.$$

Solving for ${p_{\|}}^{2}$, and ${p_{⊥}}^{2}$ in the above equations, the actual Mueller $M^{\prime }$ matrix of the wire grid polarizer is obtained. Thus, the corrected outgoing light can be determined:(32)$$\begin{array}{c} \left[\begin{array}{c} I^{\prime \prime \prime }\\ Q^{\prime \prime \prime }\\ U^{\prime \prime \prime } \end{array}\right]=\frac{1}{2}\left[\begin{array}{ccc} p_{\|}^{2}+p_{⊥}^{2} & (p_{\|}^{2}-p_{⊥}^{2})\cos2\beta & (p_{\|}^{2}-p_{⊥}^{2})\sin2\beta \\ (p_{\|}^{2}-p_{⊥}^{2})cos2\beta & \cos^{2}2\beta +2p_{\|}p_{⊥}\sin^{2}2\beta & {\left(p_{\|}-p_{⊥}\right)}^{2}\sin2\beta cos2\beta \\ (p_{\|}^{2}-p_{⊥}^{2})\sin2\beta & {\left(p_{\|}-p_{⊥}\right)}^{2}\sin2\beta cos2\beta & \sin^{2}2\beta +2p_{\|}p_{⊥}\cos^{2}2\beta \end{array}\right]\frac{I_{0}}{2}\left[\begin{array}{c} R_{s}+R_{p\;}\\ R_{s}-R_{p\;}\\ 0. \end{array}\right] \end{array}$$

The wire grid polarizer extinction ratio is shown in Figure 16. Instruments were calibrated using the above method, and the results are shown in Figure 17, Figure 18, Figure 19 and Figure 20. The black curve revised the Mueller matrix elements as the wavelength function curve; the red curve is the pre-and post-corrected deviation of the Muller matrix elements with respect to the wavelength.

### 5.3. Uncertainty Analysis and Discussion

In order to verify the accuracy of the polarization calibration correction method, an uncertainty analysis of the calibration results before and after calibration was carried out. According to the calibration method and transposition in Section 3, the main sources of uncertainty in the polarization calibration of the ultraviolet hyperspectral detector are: stability of polarized light source; turntable angle accuracy; the uniformity of the output light intensity of the UV collimator; wire grid polarizer degree of polarization; Ultraviolet Hyperspectral Detector measurement repeatability; and stray light of the Ultraviolet Hyperspectral Detector. Each source of uncertainty is analyzed below:Stability of polarized light source: The stability of the polarized light source is determined by the stability of the Hamamatsu L2479 xenon lamp. From Section 3.3, the output stability of this xenon lamp is 0.2%, so the uncertainty introduced by the stability of polarized light source is 0.2%;Turntable angle accuracy: The turntable angle error causes the polarization angle deviation, which in turn affects the polarization calibration uncertainty of the instrument. From Section 3.3, it can be seen that the turntable angle accuracy is better than 0.01°. By introducing a deviation of ±0.01° to the polarization angle of the incident spectrum, it is calculated that the deviation of the Mueller matrix element value of the instrument is less than 0.1%, so the uncertainty introduced by the turntable rotation angle accuracy is 0.1%;The uniformity of the output light intensity of the UV collimator: For the influence of the uniformity of the output light intensity of the UV collimator, please refer to the analysis in Section 4.1. When the output light of the UV collimator has a polarization degree of P, the error introduced to the measurement of the polarization characteristics of the instrument is P. Calculated from Section 5.1, the polarization response due to the UV collimator mirror is approximately 2%. Theoretically, this source of uncertainty can be completely eliminated by calibration. However, due to the influence of the collimator’s field of view of 2° (the incident on the mirror is not an ideal 0-degree parallel light) and coating defects, the residual influence after correction in Section 5.1 is estimated to be 0.8%;Wire grid polarizer degree of polarization: The manufacturer gives the typical polarization degree of a wire grid polarizer in the 290–500 nm band as better than 99%, as shown in Figure 6. In the actual test, the polarization degree of the wire grid polarizer at 290 nm measured by the device in Section 5.2 is 98.53%, and the polarization degree after 300 nm is greater than 99%. Therefore, the value of 1.5% is taken when evaluating the uncertainty introduced by the degree of polarization of the wire grid. The uncertainty source is corrected, and the residual uncertainty is estimated to be 0.5%, which is mainly caused by the uncertainty in the measurement process of the polarization degree of the wire grid polarizer;Ultraviolet Hyperspectral Detector measurement repeatability: The output signal of the instrument during polarization calibration is analyzed, and it is determined that the repeatability of the output signal of different wavelengths is about 1%;Stray light of Ultraviolet Hyperspectral Detector: The influence of stray light during polarization calibration is about 0.2% on the spectral distribution of the signal during polarization calibration and the stray light performance analysis of the instrument itself.

According to the above analysis, the comprehensive uncertainty before instrument calibration is:(33)$$\delta_{0}=\sqrt{0.2{\%}^{2}+0.1{\%}^{2}+2{\%}^{2}+1.5{\%}^{2}+1{\%}^{2}+0.2\%}=2.7\%.$$

The comprehensive uncertainty after instrument calibration is:(34)$$\delta_{0}=\sqrt{0.2{\%}^{2}+0.1{\%}^{2}+0.8{\%}^{2}+0.5{\%}^{2}+1{\%}^{2}+0.2\%}=1.4\%.$$

The specific uncertainty is shown in Table 3.

The entire system’s uncertainty was then examined to evaluate the effectiveness of the polarization properties adjustment process. The sources of uncertainty during the test, and the influence extent, are shown in the table below. The sources of uncertainty in the system comprise the stability of the polarized light source, the accuracy of the circular rotation table, the residual deviation after non-uniform correction, residual deviation after correction for polarization degree, measurement repeatability of the UV sample instrument, and the effect of stray light [39].

The above uncertainty analysis shows that the overall uncertainty after correction is 1.4%. The self-stability of the UV hyperspectral instrument is 1.0%, which would affect the measurement results to some extent. Moreover, the uncertainty of the polarization test system after correction is 1.0% [41]. As a result, the method of correction related to the influencing factors and the polarization properties measurement of the instrument suggested in this research may satisfy the requirements of polarization properties measuring accuracy for UV hyperspectral instruments.

## 6. Conclusions

This paper presents a set of polarization calibration and measurement devices for laboratory polarization calibration of UV hyperspectral instruments. The system-wide radiation transmission link of polarization was analyzed to enhance the calibration accuracy. The error was introduced by two devices: the UV collimator and the wire grid polarizer. The Muller matrix of the UV collimator regarding the polarization response was derived by analyzing its internal structure. The Mueller matrix of the wire grid polarizer was further modified by measuring the amplitude attenuation coefficient in its orthogonal direction. After correction for polarization properties, the accuracy was increased by 1.3% compared to before the correction, reaching 1.41%, providing high-precision ground calibration information for on-orbit polarization correction.

## Figures and Tables

**Figure 1 sensors-22-08542-f001:**
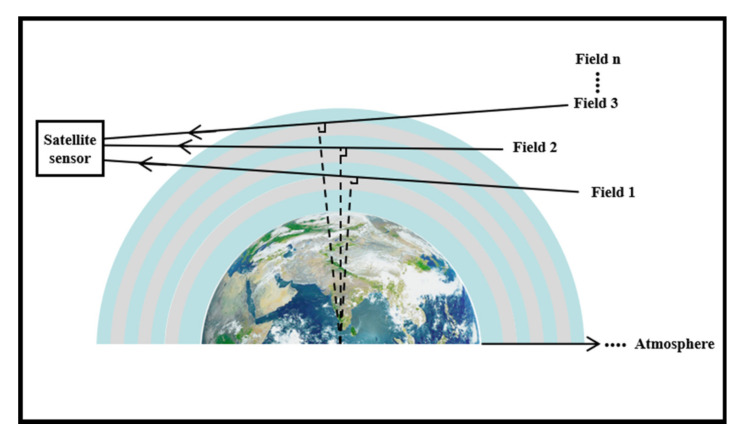
Schematic diagram of limb-scan technique.

**Figure 2 sensors-22-08542-f002:**
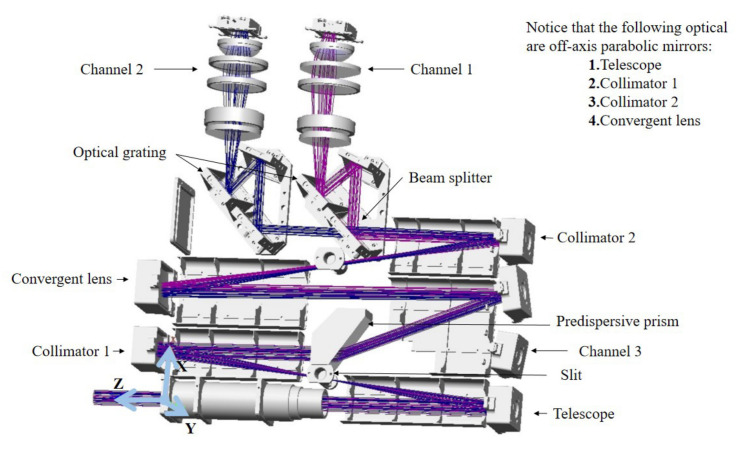
Internal structure diagram of the UV hyperspectral instrument.

**Figure 3 sensors-22-08542-f003:**
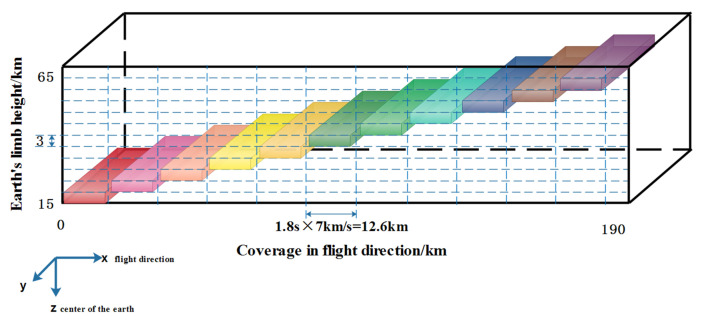
Schematic diagram of limb scanning detection.

**Figure 4 sensors-22-08542-f004:**
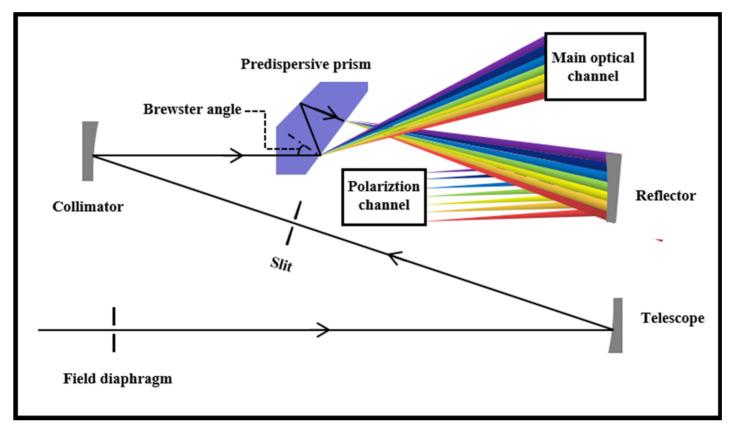
Optical structure diagram of polarization channel.

**Figure 5 sensors-22-08542-f005:**
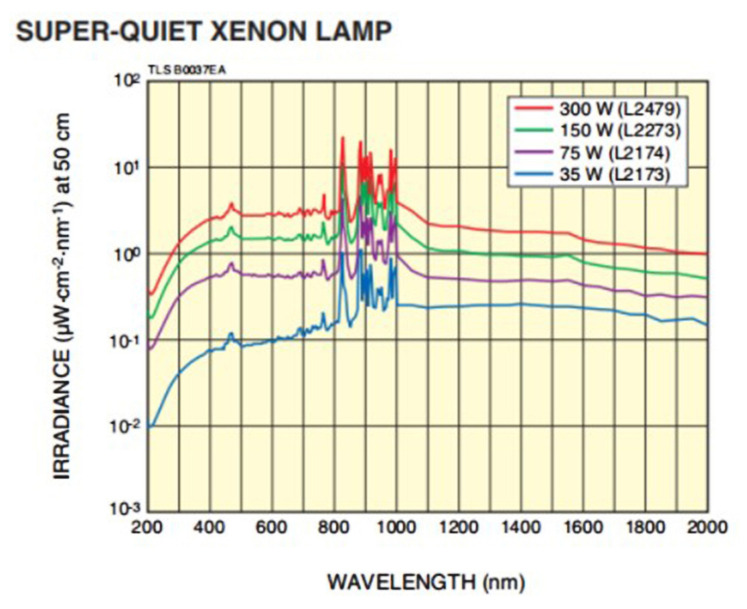
Xenon Lamp Spectral Irradiance (from SOFN INSTRUMENTS CO.).

**Figure 6 sensors-22-08542-f006:**
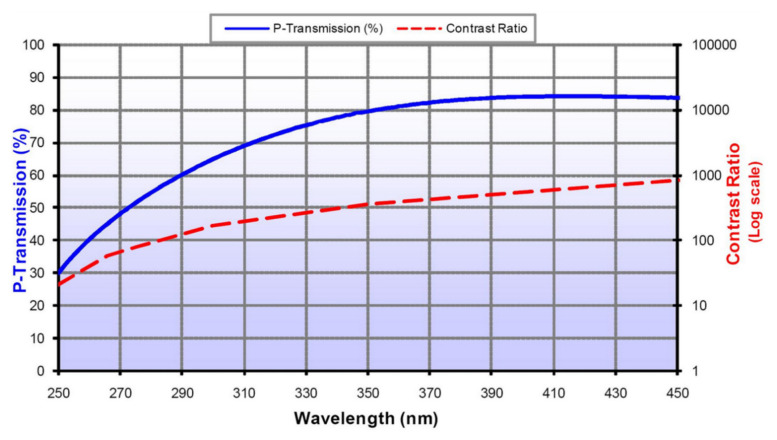
Polarization characteristic curve of wire grid polarizer (the wire grid polarizer type is 68-751, this figure is from EDMUND OPTICS Company).

**Figure 7 sensors-22-08542-f007:**
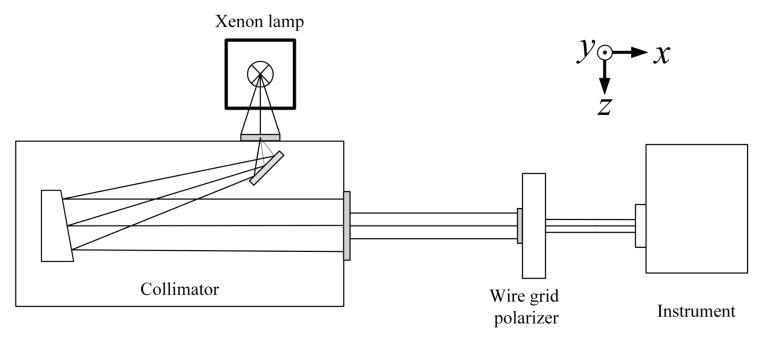
UV hyperspectral instrument’s polarization properties calibration device diagram.

**Figure 8 sensors-22-08542-f008:**
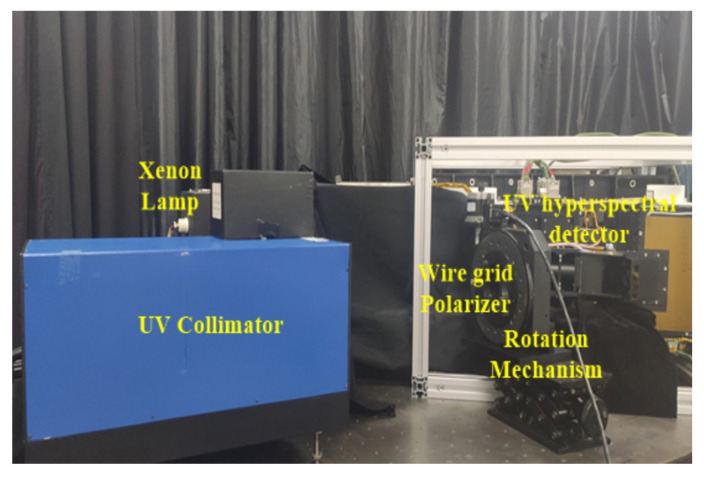
Calibration test diagram of polarization characteristics of UV Hyperspectral Detector.

**Figure 9 sensors-22-08542-f009:**
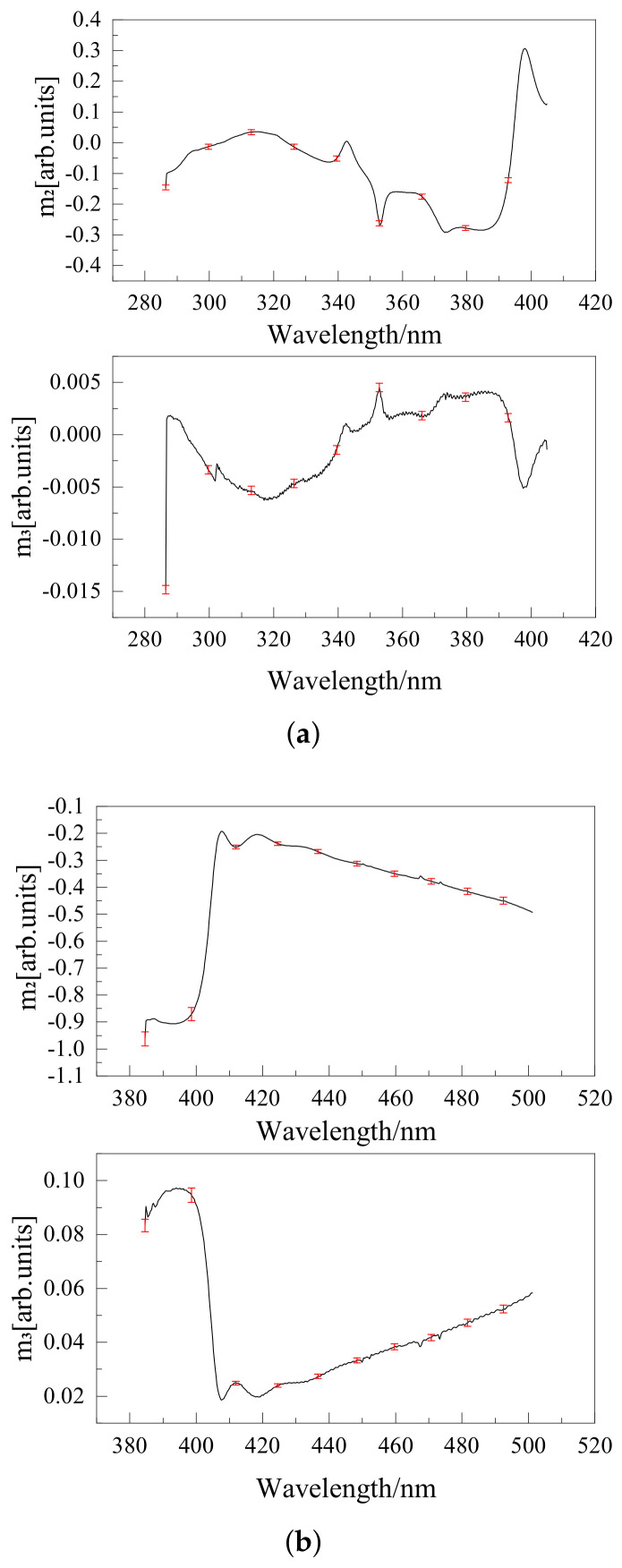
(**a**) Polarization responses $m_{2}$ and $m_{3}$ of channel 1 of the UV Hyperspectral Detector; (**b**) Polarization responses $m_{2}$ and $m_{3}$ of channel 2 of the UV Hyperspectral Detector.

**Figure 10 sensors-22-08542-f010:**
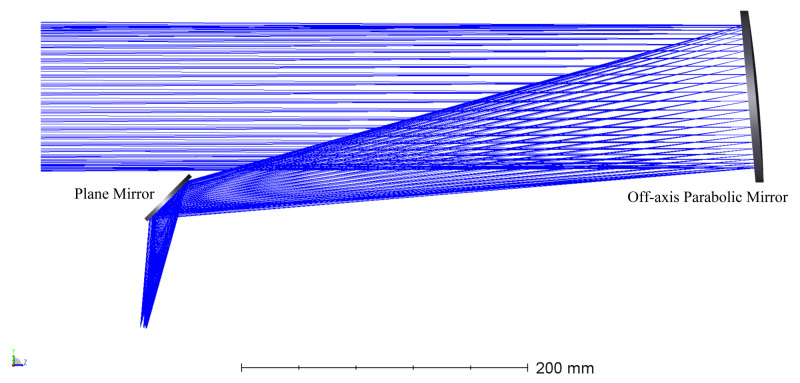
The optical structure of the collimator.

**Figure 11 sensors-22-08542-f011:**
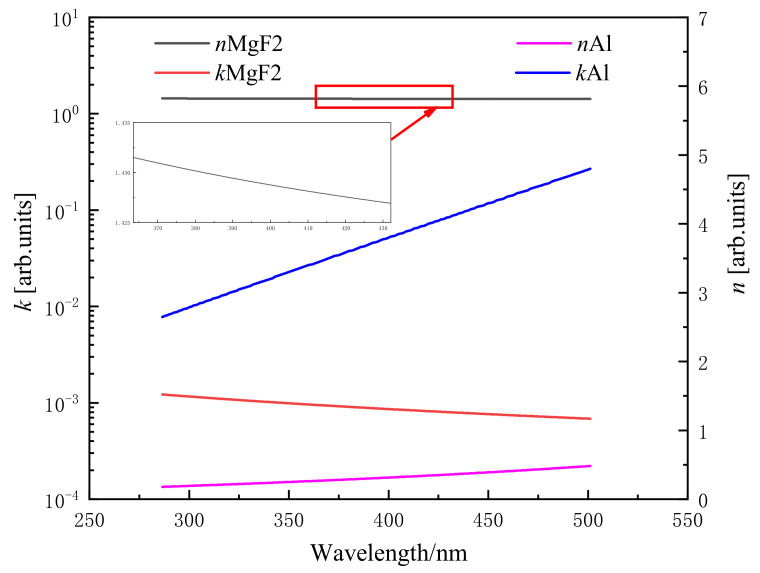
Refractive index and absorptivity of MgF2 and Al with respect to wavelength.

**Figure 12 sensors-22-08542-f012:**
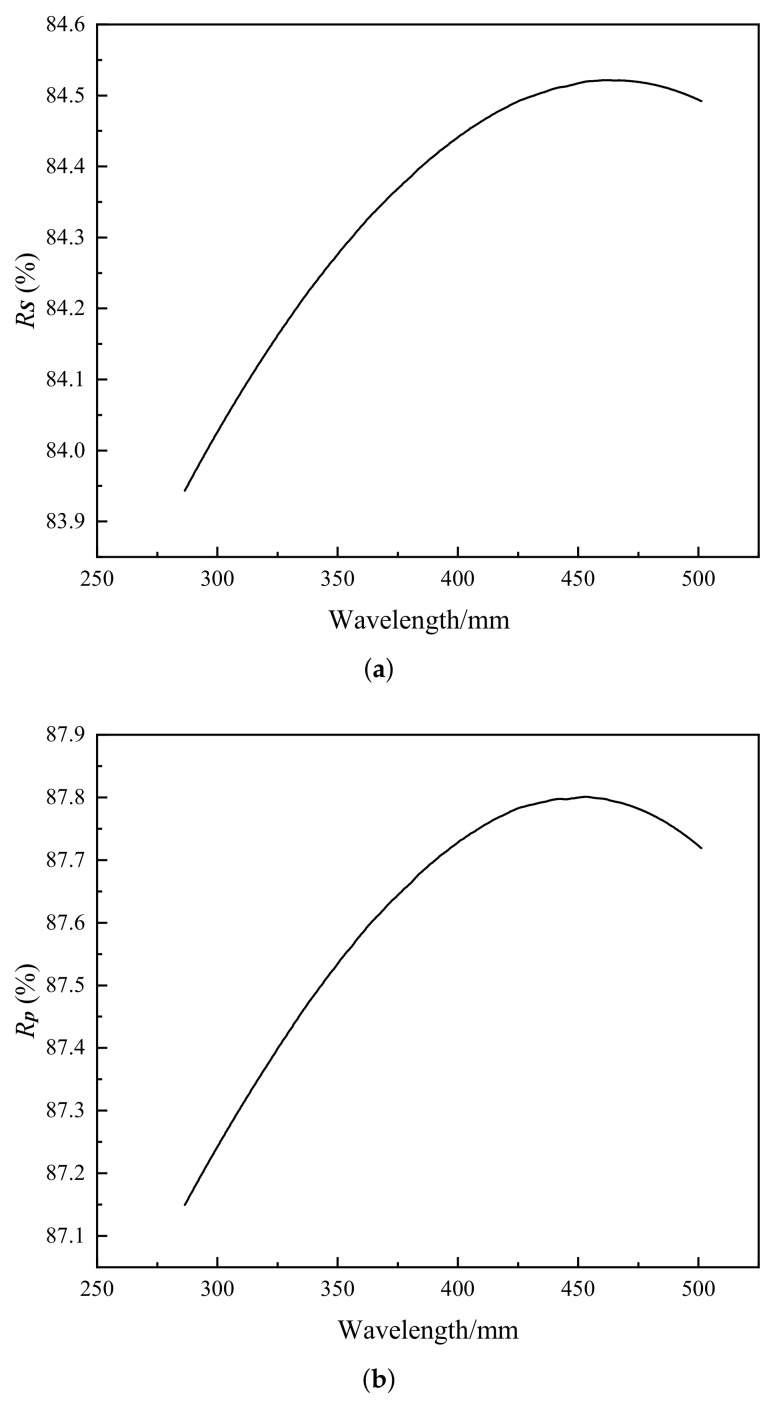
(**a**) Curve of Rs with respect to wavelength, (**b**) Curve of Rp with respect to wavelength.

**Figure 13 sensors-22-08542-f013:**
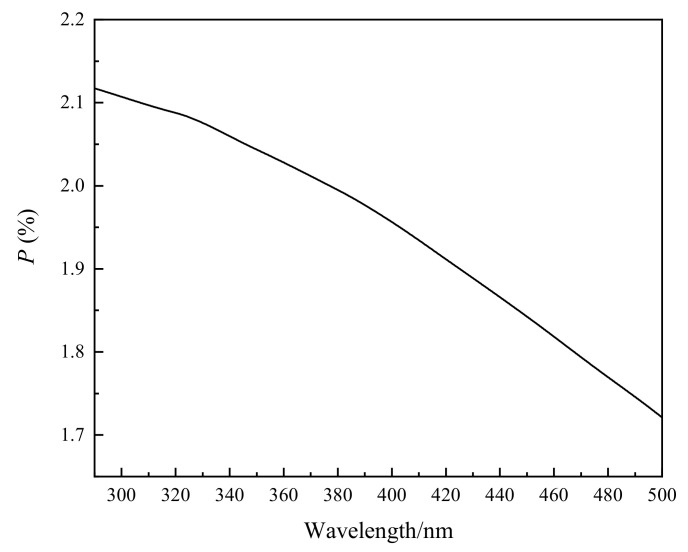
The curve of polarization degree of UV collimator as a function of wavelength.

**Figure 14 sensors-22-08542-f014:**
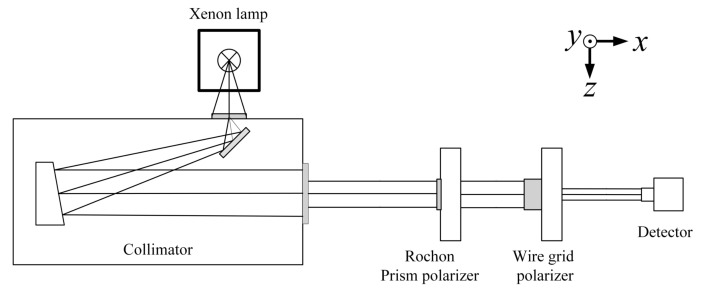
Wire grid transmission ratio test setup diagram.

**Figure 15 sensors-22-08542-f015:**
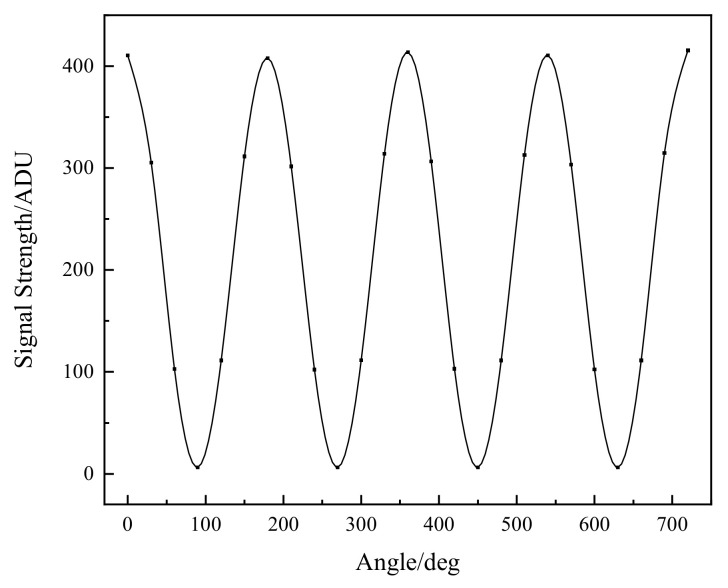
Variation curve of signal with angle at 290 nm wavelength.

**Figure 16 sensors-22-08542-f016:**
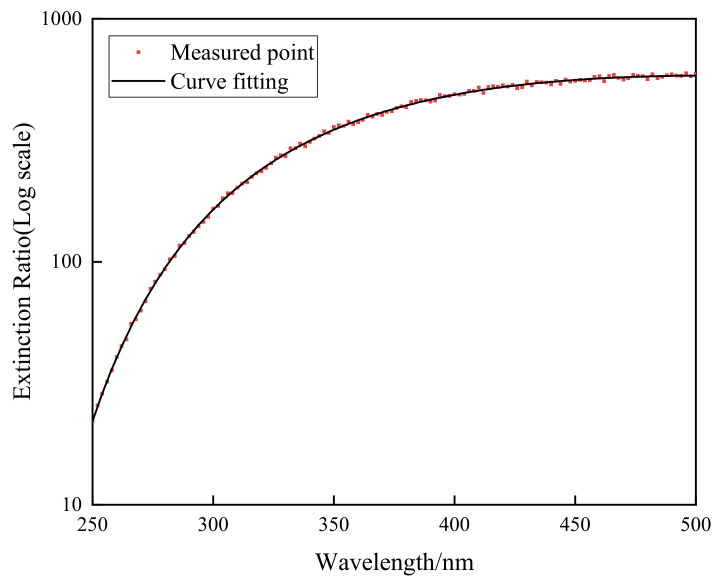
Curve of extinction ratio with respect to wavelength.

**Figure 17 sensors-22-08542-f017:**
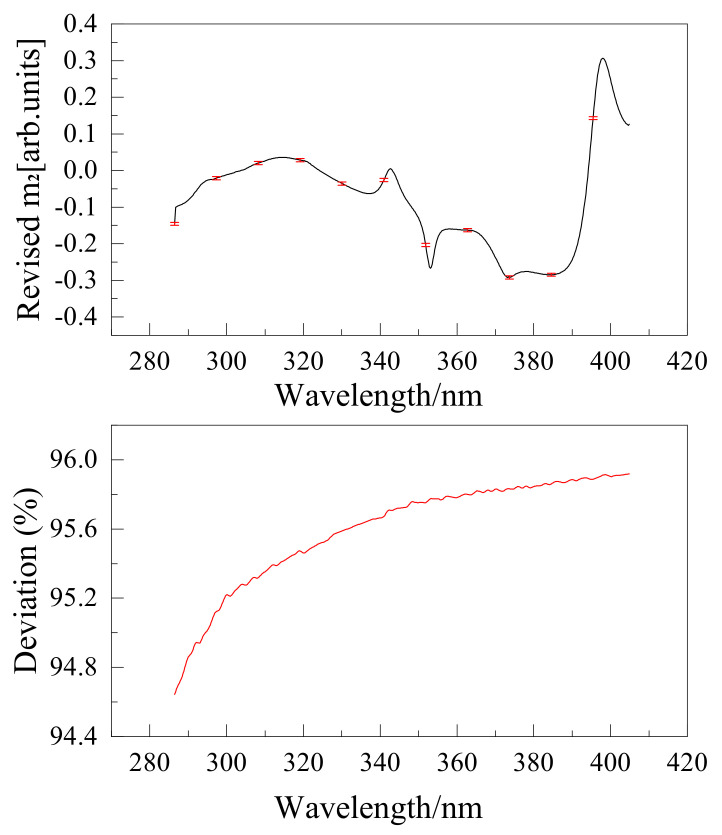
Comparison of Mueller matrix element $m_{2}$ for channel 1 before and after correction for UV hyperspectral instrument.

**Figure 18 sensors-22-08542-f018:**
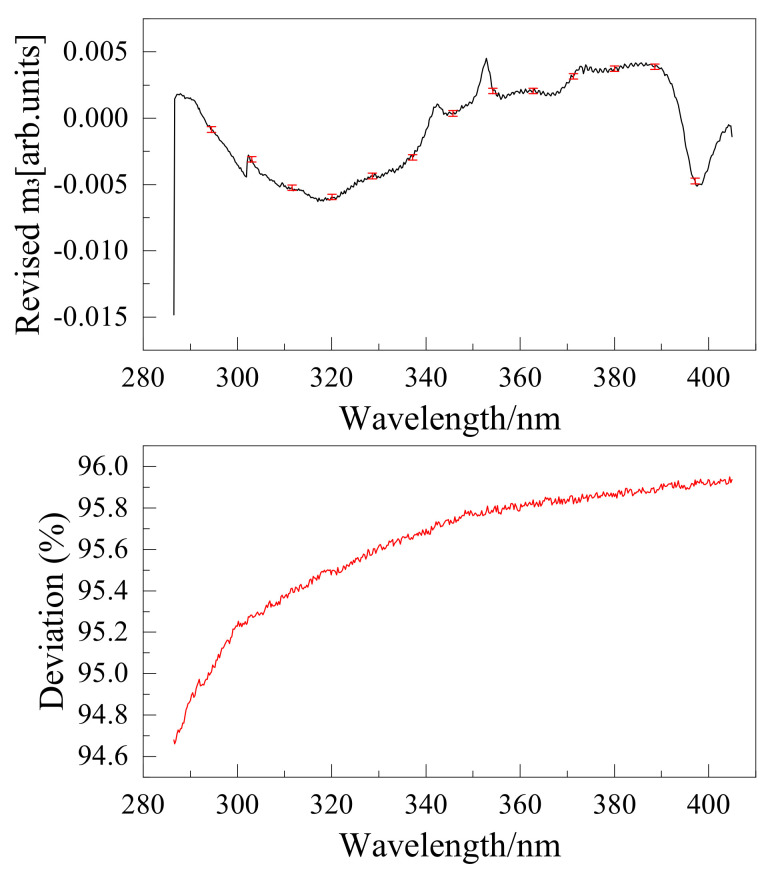
Comparison of Mueller matrix element $m_{3}$ for channel 1 before and after correction for UV hyperspectral instrument.

**Figure 19 sensors-22-08542-f019:**
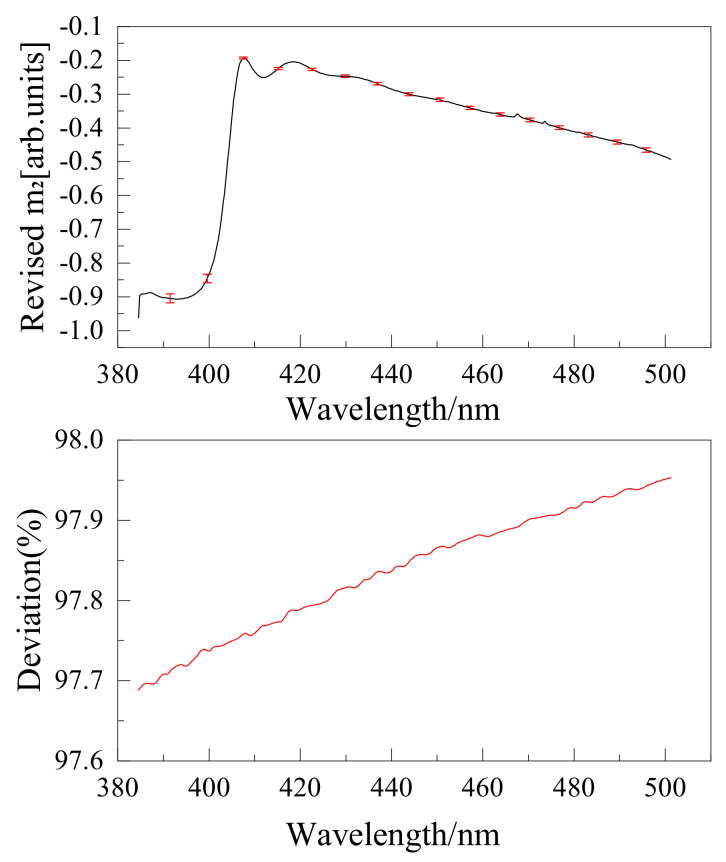
Comparison of Mueller matrix element $m_{2}$ for channel 2 before and after correction for UV hyperspectral instrument.

**Figure 20 sensors-22-08542-f020:**
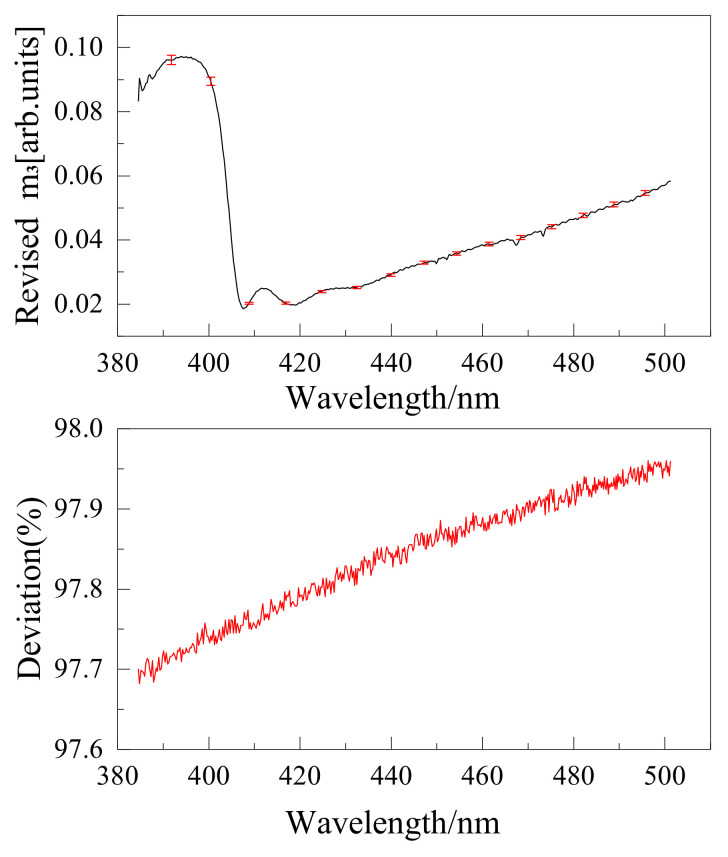
Comparison of Mueller matrix element $m_{3}$ for channel 2 before and after correction for UV hyperspectral instrument.

**Table 1 sensors-22-08542-t001:** Typical Atmospheric Ultraviolet Spectrometer Polarization Correction Scheme.

Instruments	Wavelength (nm)	Resolution (nm)	Dispersion Elements	Detector Type	Polarization Correction Scheme
TOMS	308.6–360	1.2	Grating	PMT	Depolarizer
SBUV	160–400	1.1	Grating	PMT	Depolarizer
OMPS	300–1000	1.8–40	Prism	CCD	Polarization compensator
OMI	252–740	0.4–0.6	Grating	CCD	Depolarizer
GOME	240–790	0.2–0.4	Grating	Linear array	PMD
SCIAMACHY	240–2400	0.2–0.5	Grating	Linear array	PMD

**Table 2 sensors-22-08542-t002:** Main parameters of Ultraviolet Hyperspectral Detector.

Parameter Name	Indicator Requirements
Spectral range	290–500 nm
Spectral resolution	<0.6 nm
Limb scan height	15–60 km
Spatial resolution	3 km (the height direction of the Earth’s limb)
Instantaneous field of view	1.8° * 0.045°
Radiometric accuracy	3%
Straylight	$10^{-5}$

**Table 3 sensors-22-08542-t003:** Influence extent of each source.

Sources of Uncertainty		Uncertainty before Calibration		Uncertainty after Calibration	
Polarization test system	Stability of the polarized light source	0.2%	2.5%	0.2%	1.0%
	Turntable corner accuracy	0.1%		0.1%	
	The uniformity of the output light intensity of the UV collimator	2.0%		0.8%	
	Residual error of the wire grid polarizer	1.5%		0.5%	
UV hyperspectral	Measurement repeatability	1.0%	1.0%	1.0%	1.0%
	Stray light	0.2%		0.2%	
Overall uncertainty		2.7%		1.4%	

## Data Availability

Not applicable.

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
