# Peer review of "Analysis and Correction of Polarization Response Calibration Error of Limb Atmosphere Ultraviolet Hyperspectral Detector"

_sensors, 2022, doi:10.3390/s22218542_

Round 1
Reviewer 1 Report (Previous Reviewer 2)
After the last revision the manuscript is acceptable.
Author Response
Thank you for your comments on our manuscript entitled " Analysis and Correction of Polarization Response Calibration Error of Limb Atmosphere Ultraviolet Hyperspectral Detector". Those comments are very helpful for revising and improving our paper, as well as the important guiding significance to other research. We have studied the comments carefully and made corrections which we hope meet with approval.
Reviewer 2 Report (New Reviewer)
Dear authors!
The manuscript deals with theoretical issues and experiments related to the accuracy analysis of an optoelectronic device. The results are of undoubted interest, but the article needs to be improved due to the large number of comments.
Best regards

Author Response
Please see the attachment.

This manuscript is a resubmission of an earlier submission. The following is a list of the peer review reports and author responses from that submission.
Round 1
Reviewer 1 Report
This study provides a well-constructed description of how to derive polarization assessment under lab conditions. The following are my comments:
1) Need to provide specification of the UV hyperspectral instrument and purpose of the development of the sensor including specific use of future satellite missions.
2) Is the instrument a Push broom scanner or a across-track scanner? More details need to be provided such as pixel spatial resolution, scan angle range and swath coverage.
3) Specify which angular conditions under which results of Table 1, Figs 8 and 9 are derived. If the instrument has a range of scan angle, how the results change with scan angle.
4) For Figs 8 and 9, Values of m2 are much larger than 5%. More discussion of the results and comparison with published results from similar instruments are needed to justify the current results.
Reviewer 2 Report
Polarization calibration study of UV hyperspectral instruments for atmospheric detection
The authors did present a development of a spaceborne UV-Spectrometer including a polarization channel to correct for the polarization sensitivity of the instrument. They claimed that “the calibration accuracy of the corrected polarization properties reached 1.41%, meeting the requirement of the in-orbit polarization correction accuracy.”
The topic is interesting and important for climate research for e.g. to monitor O3, NO2 and other trace gases. In this case to derive ozon profiles. A key issue to correct for the instrument polarization sensitivity is a sufficient laboratory characterization of the instrument on ground and an in-orbit polarization calibration methode. The manuscript is trying to introduce such method and claims a „high-precision ground polarization calibration for in-orbit polarization calibration.“
Overall i dont see any proof of such accuracy.
1. Table 1 gives only numbers which are not proofen by the manuscript. Where is the calculation of the resulting error?
2. Fig 1 shows a rather simplified single scattering diagram, which is not true in most cases
3. Fig 2 shows an instrument without any detailed component description. No detailed entrance optics draft is shown. A description or draft of the polarization channel is missing.
4. No real measurements of the laboratory calibration measurements are presented
5. would be nice to see some examples of in-orbit calibration measurements
6. All plots are without any error bars
7. curves have unexplained thick forms (Fig. 6 Rs and Rp)
8. Some plots have unexplained spikes, jumps or ripples (Fig.8 m³)
[124] ? „the polarization directed towards the paper (represented by the intersecting circles)“ Dont know what that means.
[129] Can you make your point more clear
[130] A plot of some measurements including error bars missing
[138] focal length 500nm, hole diameter 100nm ==> nonsense
[142] Is that true in all cases and up to what accuracy, 0.5%, 0.05% or even less
[149] If the wire grid polarizer gives only 99% of degree of polarization how can you reach an accuracy of 1.41%
[153] Is a Xenon lamp really unpolarized?
[166] Is neglecting the circular light sufficient to reach high accuracies in linear polarization?
[181] dito
[194] Where is the proof to that statement
[199] Something is wrong with the formulars, Description Ts , Tp missing
[206] If it is really unpolarized.
...
Reviewer 3 Report
I am flattered by contributing to the given journal and to this review manuscript. The work is interesting and falling within the scope of the journal. Here, authors aim to theorizing laboratory calibration due to the polarization state of ultraviolet spectrum-based hyperspectral instruments. This kind of discussion and review must be extensively referenced, as potentially provide important metrics to hyperspectral sensors, which in turn are widely applied over environmental research. However, hyperspectral sensors applied to environment mostly rely on imagery over visible and infrared spectra, as well as are not on orbital level. Considering this section subject and other considerations described below, this manuscript can be considered for publication.
General Comments
Authors could improve its introduction, by inserting more references that supports its definitions on this subtitle. Yet, the later sections demand several inclusions of references to support these definitions. Despite being a potential Sensors MPDI journal publication, several relations to environmental remote sensing applications improve its reach and fits better to this journal section.
Specific Comments
Lines 2, 3, 10, 36, 37, 41, 42, 43, 51, 257: Authors must parametrize “in-orbit” or “on-orbit”, since there is no clear difference between these terms.
Lines 18 and 19: a space between the number and the measurement unit is required.
Lines 24-25: These references (2 and 3) do not conclude to this affirmation “Most current atmospheric remote sensing instruments”, since they are about depolarizers. Hereafter this sentence, authors could insert hyperspectral sensors applicability over environmental assessment, including examples and this scenario.
Lines 28-29: Should these values be considered a sharp difference? I suggest to discuss more about the scanning mirror aperture size.
Lines 53-59: This paragraph demands several references.
Lines 70-71, 129-130,167-168, 181-182, 206-207, 211-212, 226-227: Something wrong on your document happened, line numbers are not present in these parts.
Line 136: The collimator and the lamp descriptions will improve this review.
Lines 138: a space between the number and the measurement unit is required.
Lines 140-141: Taking this sentence as example about including citations. Presenting well-established references that substantiate this Al+MgF2 film is required or improve the polarization calibration process, mainly taking into account that relies on hyperspectral sensors devoted to environmental applications.
Line 155: The Mueller matrix calibration method must be referenced.
Line 167, 206: should be a space before the bracket.
Line 181 (?): The sentence below eq. 16 should not have paragraph indent.
Line 182, 229: Check paragraph indent.
Round 2
Reviewer 1 Report
I appreciated the effort of your revision. I think the revised version of the manuscript has properly addressed my concerns. I have no further questions about the paper.
Reviewer 2 Report
Dear authors,
i did read the second version of the manuscript and it did improve a lot, but i am still not convinced that the headline and abstract are the main topic of the manuscript. In the end you are improving a calibration source and not the overall accuracy of an hyperspectral instrument, which the abstract if i understand it right suggests.
It is not clear to me why you are using a wire grid polarizer and not a Rochon prism or Glan Tompson polarizer as a calibration source.
267-268: Ripples, even/odd residuals of the offset correction?
296-297: You are writing R=R_air-MgF2 + T^2_air-MgF2 * R_MgF2-Al Is this right?
337: You are writing that you are measureing at 4 angles only. It would be much more interesting to see much more angles (0-720°) and plots, because of the allignment accuracy.
370-373: The value (error?) does change from 2% to 0.8 by estimation Why, how? Because its calculated at 5.1 Where?
376 a curve would nice
379-381: The value (error?) does change from 1,5% to 0.5 by estimation Why, how?
388: straylight 10^-5 or 0.2% difference?
392-396: how large is the accuracy? I dont think 0.01 of this calculated estimation.
396-397: This is the estimated overall accuracy of the calibration source and not of the overall system or Mueller matrix. The drift of the light source (+-0.5%) was not taking into account. How long takes a measurement with more than 4 angles. What about instrument drifts, etc...
I would suggest to concentrate on the laboratory measurements with the polarized channel of the hyperspectral instrument and the calculation of the mueller matrix and the resulting errors, which is in parts already there. But this manuscript is not sufficient in my view.
Reviewer 3 Report
I'm thankful to provide a post revision one feedback. Authors embraced my points, for which I'm pleased that those points made sense. Yet, I could be misundertood concerning references all over the paper, especially on Introduction and Materials and Methods sections.
I reaffirm to authors that several definition sentences that provide your background requires rmore eferences. In a ideal scenario, each sentence should be linked to a reference. At the very least, each paragraph on introduction and materials and methods sections must be referenced.